# 3D visualization of macromolecule synthesis

Timothy J Duerr[1], Ester Comellas[2,3], Eun Kyung Jeon[1], Johanna E Farkas[1], Marylou Joetzjer[4], Julien Garnier[4], Sandra J Shefelbine[3,5], James R Monaghan[1,6]*

[1]Department of Biology, Northeastern University, Boston, United States; [2]Department of Mathematics, Laboratori de Càlcul Numeric (LaCàN), Universitat Politècnica de Catalunya (UPC), Barcelona, Spain; [3]Department of Mechanical and Industrial Engineering, Northeastern University, Boston, United States; [4]University of Technology of Compiègne, Compiègne, France; [5]Department of Bioengineering, Northeastern University, Boston, United States; [6]Institute for Chemical Imaging of Living Systems, Northeastern University, Boston, United States

**Abstract** Measuring nascent macromolecular synthesis in vivo is key to understanding how cells and tissues progress through development and respond to external cues. Here we perform in vivo injection of alkyne- or azide-modified analogs of thymidine, uridine, methionine, and glucosamine to label nascent synthesis of DNA, RNA, protein, and glycosylation. Three-dimensional volumetric imaging of nascent macromolecule synthesis was performed in axolotl salamander tissue using whole-mount click chemistry-based fluorescent staining followed by light sheet fluorescent microscopy. We also developed an image processing pipeline for segmentation and classification of morphological regions of interest and individual cells, and we apply this pipeline to the regenerating humerus. We demonstrate our approach is sensitive to biological perturbations by measuring changes in DNA synthesis after limb denervation. This method provides a powerful means to quantitatively interrogate macromolecule synthesis in heterogenous tissues at the organ, cellular, and molecular levels of organization.

*For correspondence:
j.monaghan@northeastern.edu

**Competing interests:** The authors declare that no competing interests exist.

## Introduction

The measurement of nascent DNA, RNA, and protein synthesis in animals provides critical information on the state of cells (dividing, growing) in relation to their surrounding cells. Traditionally, radio-labeled or brominated nucleosides/amino acids are introduced to live animals, where they are incorporated into macromolecules during DNA synthesis (*Sidman et al., 1959*; *Gratzner, 1982*), transcription (*Uddin et al., 1984*; *Wansink et al., 1993*), and translation (*Garlick et al., 1980*). Performing such approaches have allowed for the characterization of cells actively undergoing macromolecular synthesis as well as quantification of synthesis rates, which facilitates the study of cell behavior during tissue remodeling, proliferation, stress, or disease (*Yoshizawa et al., 1997*; *Rombouts et al., 2016*; *O'Brien and Lis, 1993*; *Rose et al., 1975*).

In the past decade, bio-orthogonal macromolecule precursor analogs including 5-ethynyl-2′-deox-yuridine (EdU), 5-ethynyl-uridine (5-EU), and L-azidohomoalanine (AHA) have become commercially available for analysis of DNA synthesis, transcription, and translation, respectively. After injection of macromolecule precursor analogs into animals, precursors can later be detected in nascent DNA, RNA, and protein macromolecules with fluorescently labeled azides or alkynes through highly selective copper-catalyzed azide-alkyne cycloaddition ('click') chemistry (*Kolb et al., 2001*; *Salic and Mitchison, 2008*; *Best, 2009*). These powerful new analogs provide an alternative to the use of dangerous isotopes and the challenges associated with brominated precursors such as the requirement

**eLife digest** Cells often respond to changes in their environment by producing new molecules and building new cell components, such as proteins, which perform most tasks in the cell, or DNA and RNA, which carry genetic information. Complex tissues – such as limbs, which are made up of muscles, tendons, bones and cartilage – are difficult to see through, so studying when and where cells in these tissues produce different types of molecules is challenging. New approaches combining advanced three-dimensional microscopy and fluorescent labelling of molecules could provide a way to study these processes within whole animal tissues. One application for this is studying how salamanders regrow lost limbs.

When salamanders such as axolotls regrow a limb, some cells in the limb stump form a group called the blastema. The blastema contains cells that are specialized to different purposes. Each cell in the blastema produces many new proteins as well as new DNA and RNA molecules. Fluorescently labeling particular molecules and taking images of the regenerating limb at different times can help to reveal how these new molecules control and coordinate limb regrowth.

Duerr et al. developed a three-dimensional microscopy technique to study the production of new molecules in regenerating axolotl limbs. The method labeled molecules of different types with fluorescent markers. As a result, new proteins, RNA and DNA glowed under different colored lights. Duerr et al. used their method to show that nerve damage, which hinders limb regrowth in salamanders, reduces DNA production in the blastema.

There are many possible applications of this microscopy method. Since the technique allows the spatial arrangement of the cells and molecules studied to be preserved, it makes it possible to investigate which molecules each cell is making and how they interact across a tissue. Not only does the technique have the potential to reveal much more about limb regrowth at all stages, but the fluorescent markers used can also be easily adapted to many other applications.

of large secondary antibodies (~150 kd) and harsh tissue retrieval methods that limit their use in whole tissue samples.

Advantages of the click-chemistry labeling approach include the inert nature of macromolecule precursor analogs that have minimal impact on the animal, the small size of fluorescently labeled alkynes and azides, and the high selectivity of click-chemistry. These advantages have enabled whole-mount fluorescent labeling of DNA synthesis (*Salic and Mitchison, 2008*), RNA transcription (*Jao and Salic, 2008*), protein translation (*Hinz et al., 2012*), and glycans (*Sawa et al., 2006*; *Laughlin and Bertozzi, 2009*) in animals. These pioneering proof-of-principle experiments have demonstrated that imaging macromolecular synthesis is possible, but the fact that most model organisms are large, optically opaque, and consist of heterogenous tissues has made it a challenge to image biological phenomena in deep tissues. Challenges such as photon penetration, differences in refractive indices among different cellular components, light-induced photo-damage, and background fluorescence have limited the use of whole-mount imaging of macromolecular synthesis.

Advances in light sheet fluorescence microscopes (LSFMs) have recently enabled the imaging of large biological specimens from millimeters to several centimeters in size (*Power and Huisken, 2017*; *Dodt et al., 2007*; *Keller and Dodt, 2012*). LSFMs have been utilized for volumetric imaging of many varieties including visualization of mRNA in whole-mount fluorescence in situ hybridization experiments (*Mano et al., 2018*) and in vivo interrogation of deep tissue dynamics in transgenic reporter animals (*Tomer et al., 2012*), among others (*Power and Huisken, 2017*). Clearing methods like CLARITY (*Chung et al., 2013*), CUBIC (*Susaki et al., 2014*), and 3DISCO (*Ertürk et al., 2012*) have further enabled volumetric imaging by decreasing refractive index mismatches and tissue clearing to advance large specimen imaging even further. Together, the rise of three-dimensional imaging, new staining techniques, and tissue clearing has demanded new means for cell counting, segmentation, and fluorescence quantification.

Here we present a click-chemistry based method to visualize DNA synthesis, transcription, translation, and protein glycosylation in whole-mount samples using LSFM (*Figure 1A*). We demonstrate the utility of this technique by imaging macromolecular synthesis in the regenerating axolotl

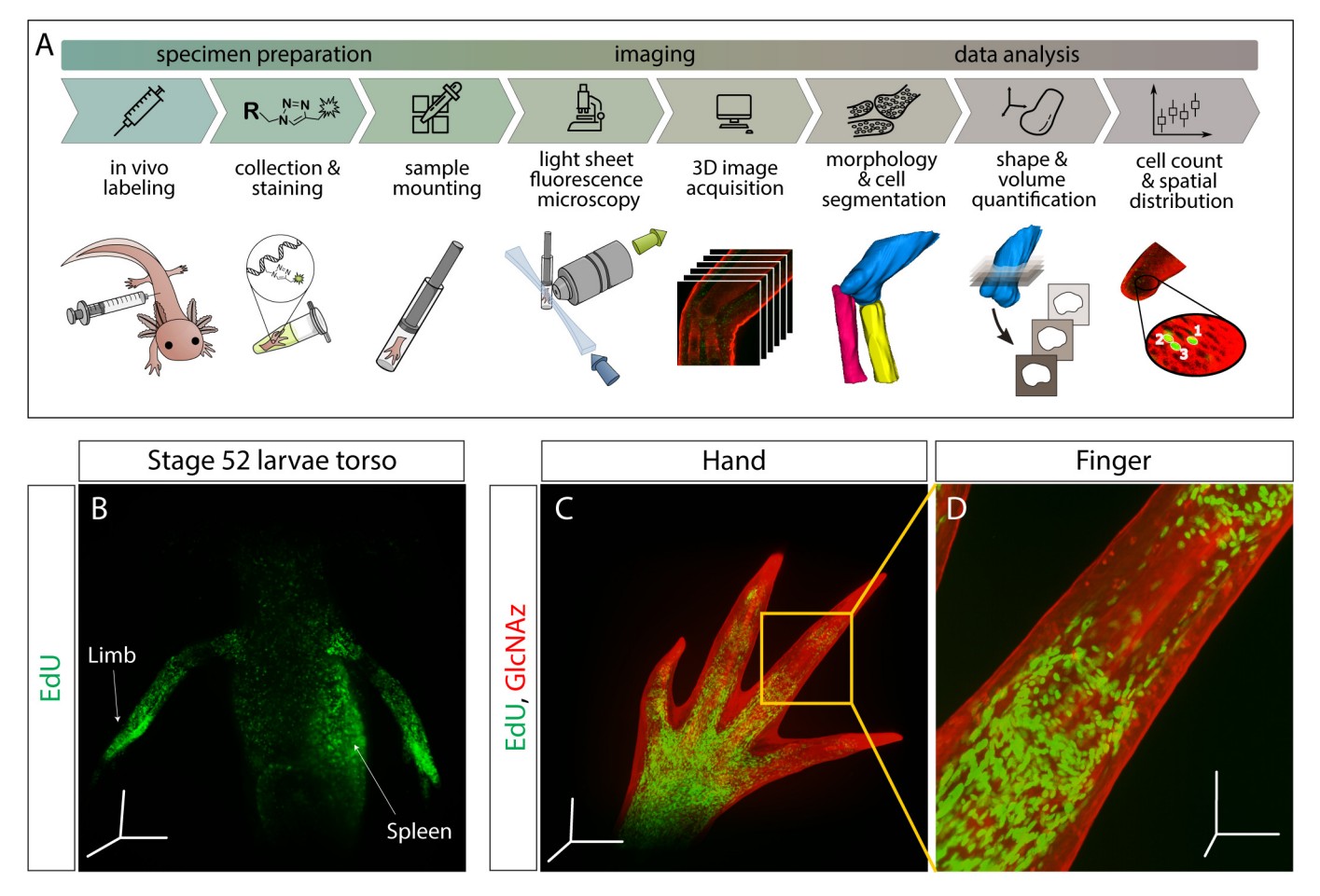

**Figure 1.** Outline of staining/analysis pipeline and exemplary images. (**A**) Overview of entire sample preparation, imaging, and data analysis pipeline. (**B–D**) Once macromolecules are labeled in vivo, synthesis can be visualized throughout the injected animal. Here we show DNA synthesis (EdU) in the torso of a stage 52 larvae (**B**) and both DNA synthesis and protein glycosylation (GlcNAz) in the hand (**C**) and finger (**D**). For panels B-D, animals were pulsed with the corresponding macromolecule analog(s) for 3 hr. Images from panels B and C were uncleared and imaged at 5× magnification. Image from panel D was also uncleared and imaged at 10× magnification. Scale bars for panels B and C = 600 µm for each axis. Scale bars for panel D = 200 µm for each axis.

salamander limb. Following limb amputation, the axolotl regenerates its limb by generating a mass of proliferating cells at the limb stump called a blastema (*Stocum, 2017*). The blastema is an ideal environment to test our method because it is an accessible, heterogenous tissue that increases DNA synthesis, transcriptional output, and translation rates compared to uninjured tissue. Furthermore, DNA, RNA, and protein synthesis decrease after denervation of the regenerating limb (*Singer and Caston, 1972*; *Dresden, 1969*).

We use the regenerating limb to demonstrate that two click-it ready precursors can be administered and subsequently visualized simultaneously in a single sample. We also show that optical resolution of images can be improved with the clearing agent 2,2'-thiodiethanol (TDE), and that this method works in a number of tissues. We outline an image analysis pipeline for three dimensional (3D) morphology segmentation, cell counting, and fluorescence quantification of stained tissues. We apply this pipeline to the regenerating humerus to demonstrate the multiscale quantitative analysis capabilities of our method. Finally, we show that our method is sensitive enough to detect and quantify changes in DNA synthesis rates in whole-mount innervated and denervated regenerating limbs. Taken together, our method provides a unique approach to simultaneously interrogate cell state at the organ, cellular, and molecular levels of organization.

## Results

## Whole-mount, click-it based visualization of macromolecule synthesis

To visualize macromolecule synthesis, we injected click-it compatible monomer analogs (*Table 1*) intraperitonally 3 hr before sample collection. During this time, analogs metabolically incorporated into nascently synthesized macromolecules, resulting in in vivo labeling of macromolecule synthesis. Our group has found that collection of tissue after 3 hr provides sufficiently strong metabolic labeling of DNA synthesis (EdU), transcription (EU), translation (AHA), and protein glycosylation (GlcNAz). These labeled macromolecules contain either azide- or alkyne-modified monomers that can be detected with click-it compatible fluorescent molecules, enabling imaging of nascent macromolecules in whole-mount tissues with LSFM (*Figure 1B–C*). We show that our whole-mount method can be used at the organismal level to visualize the whole torso of a stage 52 axolotl larvae, where we observe proliferating cells in the developing limbs and hematopoiesis within the spleen (*Figure 1B*). To demonstrate the multiplexing capabilities of the approach, modified monomer analogs with disparate functional groups (EdU/AHA, 5-EU/AHA, EdU/GlcNAz, 5-EU/GlcNAz) were co-injected and visualized with LSFM (*Figure 2A–D*, *Videos 1–3*. For color blind accessible images, see *Figure 2— figure supplement 1*). whole-mount samples were comparable to 2D longitudinal tissue sections of the same stains (*Figure 2I–L*), showing that our method generates similar results in both whole-mount and tissue sections (*Figure 2—figure supplement 2*) with subcellular resolution (*Figure 2— figure supplement 3*). Furthermore, the number of EdU$^+$ cells per blastema area in a single Z slice obtained by LSFM (EdU-AHA: 1954 cells/mm$^2$, EdU-GlcNAz: 1458 cells/mm$^2$) is similar to that of a tissue section obtained by confocal microscopy (EdU-AHA: 1272 cells/mm$^2$, EdU-GlcNAz: 1349 cells/mm$^2$). We finally demonstrate the specificity of GlcNAz incorporation by pretreating a GlcNAz-specific antibody on tissue sections collected from GlcNAz injected animals. By doing so, the subsequent click-it reaction was prevented due to the antibodies likely sterically hindering alkyne fluorophores from accessing the azide functional groups (*Figure 2—figure supplement 4*).

An advantage of our method is that staining whole-mount tissues eliminates the need for sectioning, reducing the potential inconsistencies that arise as a result of the sectioning process (uneven tissue, different cutting planes, etc.). Additionally, traditional methods of obtaining 3D images of thick tissues with confocal microscopy are impractically slow when imaging hundreds of images in a single stack. LSFM allows for more rapid imaging of whole samples, requiring only minutes to image each sample. However, several considerations exist when imaging whole tissue in 3D with LSFM. Stain penetrates slower in whole 3D tissues compared to 10–20 µm thick tissue sections, requiring longer staining times. An advantage of our method is that the click reaction requires molecularly diminutive reagents that readily pass through cell and nuclear membranes, ensuring stain penetration in the center of dense tissues. Different refractive indices between disparate cellular components and the imaging media cause light to scatter, which can reduce the resolution and brightness of 3D images (*Fadero and Maddox, 2017*). To improve image resolution and light penetration, overnight refractive index matching with 67% 2,2'-thiodiethanol (TDE) can sufficiently clear axolotl limbs for imaging with LSFM. TDE was chosen due to its simplicity, cost, safety, and compatibility with common imaging modalities. This clearing method rapidly and effectively improves the signal-to-noise ratio of stained samples compared to imaging in PBS (*Figure 2—figure supplement 5*). Tissue morphology was minimally disturbed after clearing, with only mild shrinkage of the blastema epithelium observed. With careful attention to these challenges, our method provides a means to obtain high-

**Table 1.** Monomer analogs used to demonstrate the whole-mount visualization method.

| Name | Macromolecule analog | Biological process | Click-it modification | Reference |
|---|---|---|---|---|
| 5-ethynyl-2'-deoxyuridine (EdU) | Thymidine | DNA synthesis | Alkyne | *Salic and Mitchison, 2008* |
| 5-Ethynyl Uridine (5-EU) | Uracil | Transcription | Alkyne | *Jao and Salic, 2008* |
| L-Azidohomoalanine (AHA) | Methionine | Translation | Azide | *Wang et al., 2017* |
| Azide-modified glucosamine (GlcNAz) | Glucoseamine | Protein glycosylation | Azide | *Laughlin et al., 2006* |

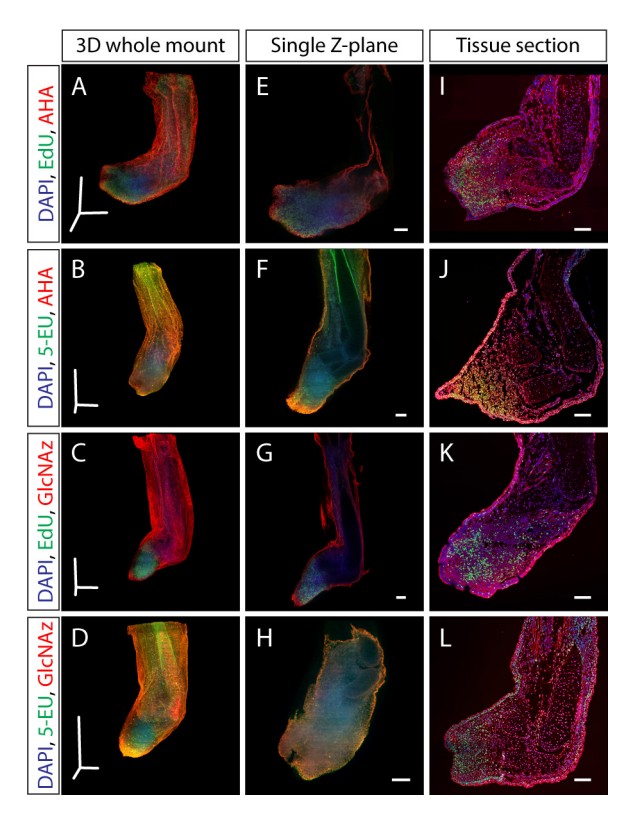

**Figure 2.** Dual staining of macromolecule synthesis in whole-mount imaging. (**A–D**) Stitched and fused 3D reconstruction of 13 dpa blastemas stained for multiple macromolecules obtained by LSFM. (**E–H**) Single Z-plane from A-D that represents the entirety of the blastema. (**I–L**) Tissue section from identically treated limbs as A-H showing similar macromolecule staining patterns, indicating that the whole-mount staining method does not alter macromolecule synthesis staining patterns. Scale bars for panels A-D = 600 μm for each axis. Scale bars for panels E-L = 200 μm.

The online version of this article includes the following figure supplement(s) for figure 2:

**Figure supplement 1.** Color blind friendly images from *Figure 2*.
**Figure supplement 2.** Single staining of macromolecule synthesis in whole-mount imaging.
**Figure supplement 3.** Subcellular resolution obtained with LSFM.
**Figure supplement 4.** Specificity of GlcNAz staining.
**Figure supplement 5.** Comparison of imaging in PBS and 67%TDE.

quality 3D images from tissues 1 mm in depth in less than 10 min per sample with clear, consistent staining (*Figure 2A–H*, *Figure 2—figure supplement 3*).

## 3D, multiscale quantitative analysis of the regenerating humerus

To demonstrate that our whole-mount click-it method can obtain quantifiable data on the organ, cellular, and molecular levels of organization, we applied our technique to the regenerating humerus. After regenerating for 35 d, axolotls with mid-humeral amputations were injected with EdU/AHA to identify cells within the humerus undergoing DNA synthesis (EdU) and protein translation (AHA). We observed EdU staining in chondrocytes distal to the amputation plane and AHA staining in the humerus perichondrium (*Figure 3A*). We outline a multiscale, quantitative pipeline that leverages the staining patterns of these macromolecules for analysis of 3D humerus morphology and 3D macromolecule synthesis. This workflow combines available plugins in Fiji (*Schindelin et al., 2012*) and scripts developed in Fiji and Matlab (*The MathWorks, 2019*) for the data analysis process (see supplementary information for a detailed description).

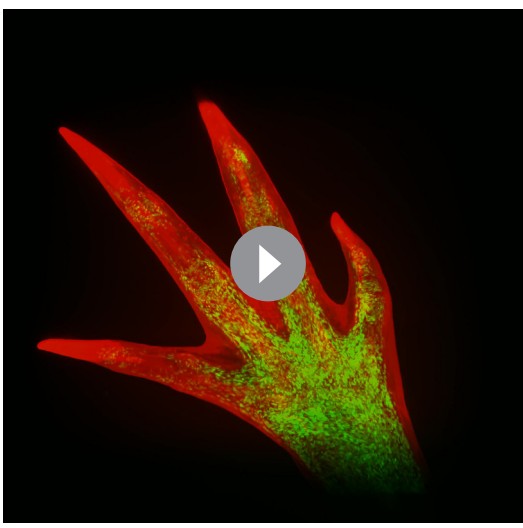

**Video 1.** Rotating axolotl hand stained for EdU (green) and GlcNAz (red).
https://elifesciences.org/articles/60354#video1

For 3D organ level analysis of macromolecule synthesis, AHA staining provided an adequate outline of the humerus, allowing us to segment its 3D morphology (*Figure 3A–B*). We quantified organ shape by assessing cross-sectional area (*Figure 4A'*) and circularity (*Figure 4A''*) of the segmented humerus along the proximodistal axis with BoneJ (*Doube et al., 2010*; *Figure 4A*). These measurements of 3D organ size can be obtained with other methods such as microCT and focused ion beam scanning electron microscopy (FIB-SEM). However, microCT is unable to image every tissue due to stain limitations and uses hazardous radiation, while FIB-SEM can capture 3D surface topography but not the entire organ morphology. Our method has the capability to fully image the 3D structure of entire organs (size permitting) without the need for radiation while simultaneously capturing information from both the cellular and molecular levels of organization.

On the cellular level, we segmented the 3D morphology of proliferating chondrocytes based on EdU staining using the Trainable Weka Segmentation 3D plugin (*Arganda-Carreras et al., 2017*; *Figure 4B''*). With this segmentation, we identify highly condensed regions of cells undergoing rapid rates of DNA synthesis (*Figure 4B*). From these data, we observe cells synthesizing DNA most abundantly distal to the plane of amputation, as expected for dividing chondrocytes. Traditionally, cell quantification as such is conducted on 2D tissue sections. In heterogenous tissues, however, cells are distributed non-uniformly; 2D sampling may not accurately capture the cellular distribution and cannot be used to determine cell volume or shape. Our whole-mount staining method allows quantification of cells within an entire 3D tissue, resulting in a more accurate assessment of cell density within heterogenous tissues.

To demonstrate the quantitative molecular analysis of 3D macromolecule synthesis, we assessed the staining intensity of EdU in slices along the proximodistal axis of the regenerating humerus (*Figure 4C*). EdU intensity represents the rate at which a cell undergoes DNA synthesis, which is one of the first steps of cell division and proliferation. Proliferating cells rapidly synthesize DNA, which enables these cells to integrate EdU into nascent DNA strands. This provides ample opportunities for covalent linkage of fluorescent molecules to the DNA strand. Thus, higher pixel values within EdU$^+$ cells are representative of more DNA synthesis in the cell. These data show that EdU intensity is strongest in the regions distal to the amputation plane of the humerus (*Figure 4C'*), providing a quantitative measure to assess macromolecule synthesis on the molecular level. Within a tissue,

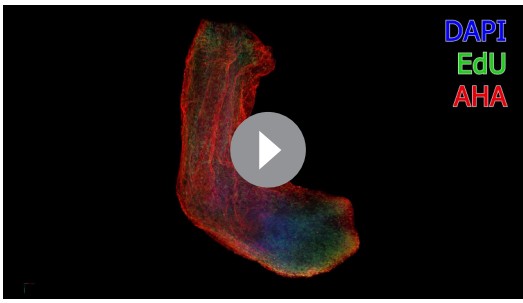

**Video 2.** Rotating axolotl limb stained for EdU (green), AHA (red), and DAPI (blue).
https://elifesciences.org/articles/60354#video2

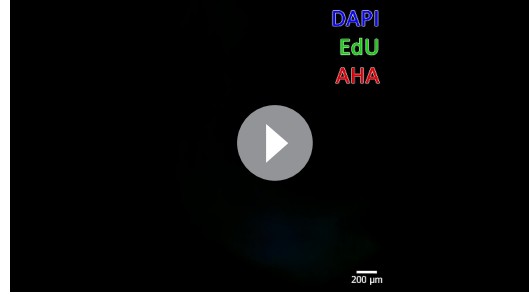

**Video 3.** Scroll through of Z-stack from *Video 2*.
https://elifesciences.org/articles/60354#video3

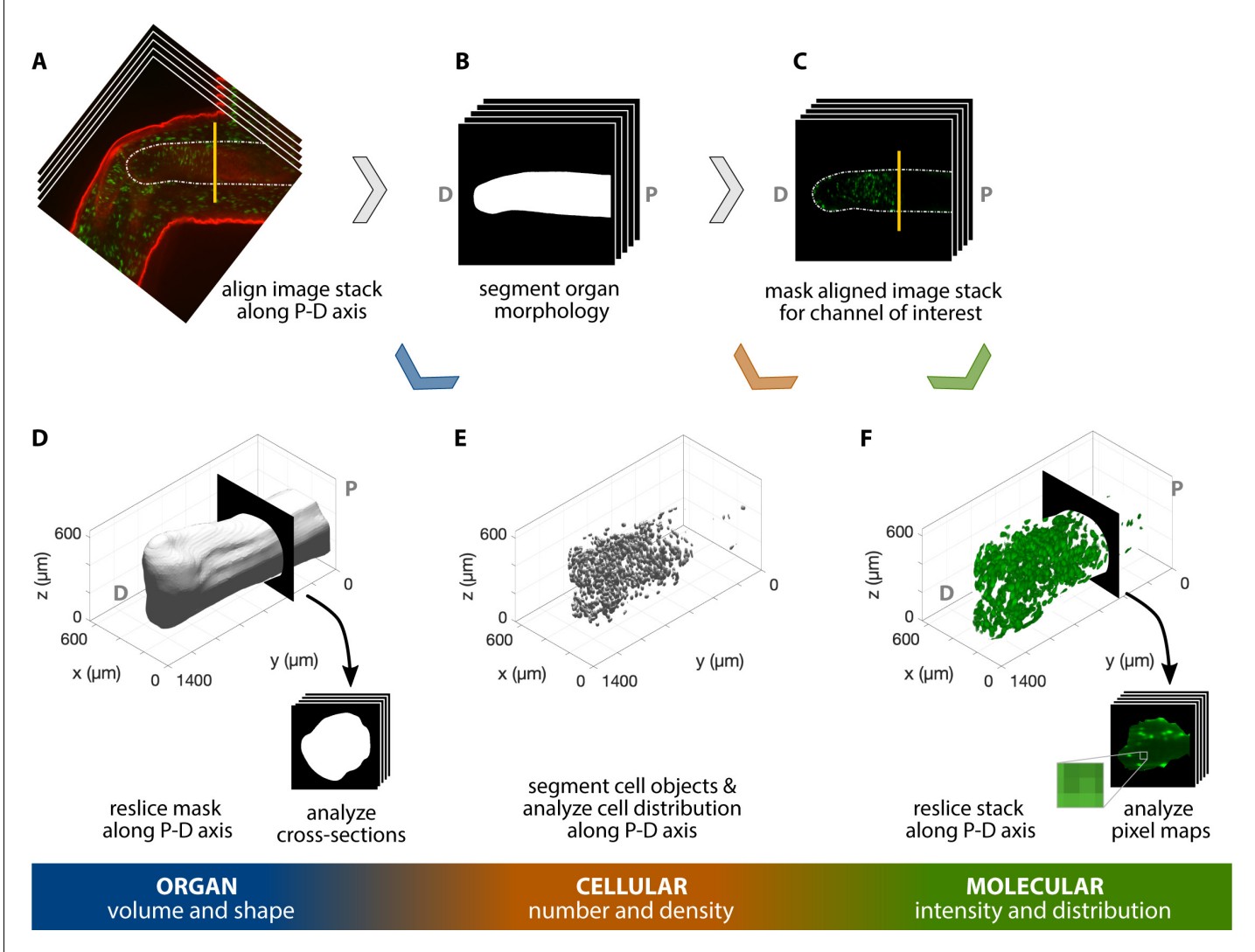

**Figure 3.** Workflow for 3D, multiscale analysis of the regenerating axolotl humerus. Multiscale analysis of a 35 dpa regenerating axolotl humerus, stained for AHA (red) and EdU (green). The humerus in the image stack was (**A**) aligned along the proximodistal (**P–D**) axis and (**B**) its morphology was segmented. The resulting mask was used to analyze the organ volume and shape by (**D**) reslicing it along the P-D axis and studying the cross-sections obtained. (**C**) The segmented morphology in B was used to mask the green channel for cellular- and molecular-level analyses. (**E**) Cells in the humerus were segmented and their spatial distribution was analyzed to obtain cellular number and density. (**F**) The masked image stack in C was resliced along the P-D axis and the pixel maps of the cross-sections were used to characterize the molecular intensity and distribution within the humerus. The vertical yellow line in A and C indicates the plane of amputation.

cells synthesize macromolecules heterogeneously, reflected by different fluorescence values between cells. Thus, quantifying macromolecule synthesis based on the presence/absence of signal instead of fluorescence does not account for variability in macromolecule synthesis rates among cells. To compound this issue, quantifying fluorescence in tissue sections only represents the rate of macromolecule synthesis from a fraction of cells in larger, heterogenous tissues. Our method provides a means to capture this molecular heterogeneity in 3D samples, allowing us to observe whole 3D regions in the regenerating humerus that synthesize DNA more rapidly than others, which further demonstrates the utility of our whole-mount click staining method.

Taken together, these results demonstrate that our method can provide quantifiable data on the organ, cellular, and molecular levels of organization. This highlights the novelty of our method, as we have not found previous examples of multiscale analysis as outlined here. We foresee this multiscale, quantitative analysis having broad applications in the examination of dynamic cell processes in

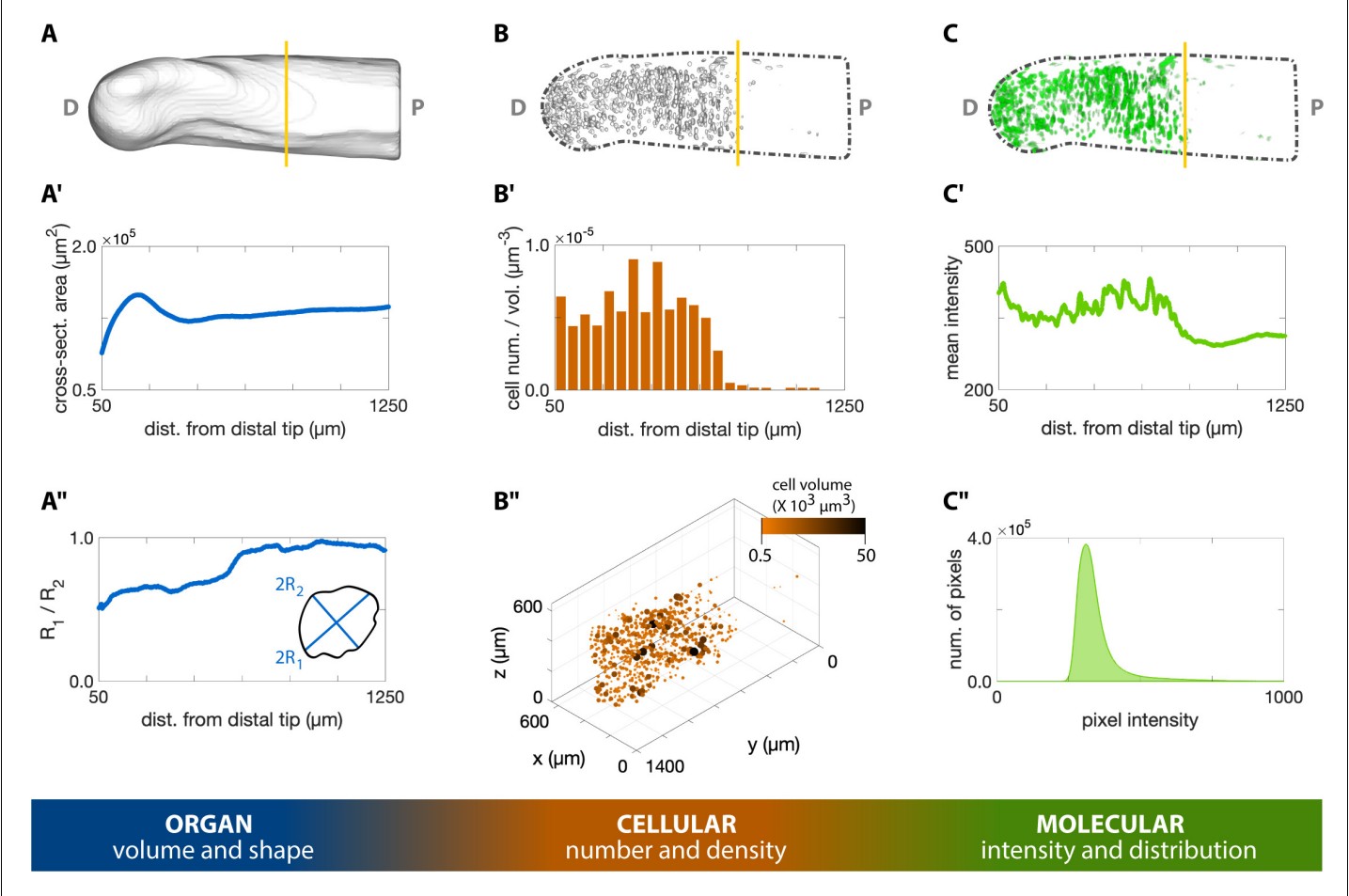

**Figure 4.** 3D quantification across scales of a regenerating axolotl humerus. (**A**) The cross-sections of the humerus in **Figure 3D** were analyzed with the Fiji plugin BoneJ to quantify humerus shape and volume. (**A'**) The cross-sectional area along the proximodistal (**P–D**) axis provides a measure of volume distribution along the humerus. (**A"**) The ratio of the maximum chord length from the minor axis (2 $_{R1}$) with respect to the maximum chord length from the major axis (2 $_{R2}$) provides a measure of cross-sectional circularity in the humerus. Values closer to 1.0 in the proximal side indicate a more circular cross-section in this zone. (**B**) The Fiji plugin Trainable Weka Segmentation 3D and 3D Objects Counter were used in the cellular analysis of proliferating chondrocytes illustrated in **Figure 3E**. (**B'**) The number of EdU$^+$ cells within a 50 µm slice along the P-D axis was divided by the slice volume to obtain a density-like measure. (**B"**) The center of mass of each cell was plotted in 3D, with point size and color proportional to the segmented cell volume. (**C**) The molecular intensity and distribution were analyzed based on the resliced pixel intensity maps of the masked green channel in **Figure 3F**. (**C'**) Mean intensity of each slice perpendicular to the P-D axis. (**C"**) The histogram of the EdU staining in the humerus can provide a measure of the DNA synthesis rate. The vertical yellow line in the top row images indicates the plane of amputation.

3D, such as in cancer metabolism and mammalian neurogenesis or other fields where macromolecule synthesis is traditionally studied in tissue sections.

## 3D, molecular analysis of biological perturbations

To demonstrate that our method is sensitive enough to detect subtle changes in macromolecule synthesis in vivo, we quantified the difference in EdU intensity between limbs regenerating with and without a nerve supply. Blastema cells are thought to have a cell cycle length of 40–50 hr, with S phase approximately 30 hr (**Tassava et al., 1987**). In newts, it has been shown that amino acid, RNA, and DNA analog incorporation decreases approximately 30% by 24 hr after denervation (**Singer and Caston, 1972**). Based upon these estimates, we amputated both forelimbs at the mid-humerus and denervated the left limb at the brachial plexus 24 hr before collection. This timepoint for denervation was chosen because it should be sufficiently long to have an impact on DNA, RNA, and protein synthesis rates. At 6, 9, 12, 15, 18, and 25 d post-amputation (dpa) animals were pulsed with EdU for 3 hr before collection to label proliferating blastema cells (**Figure 5A**). We chose 3 hr of EdU

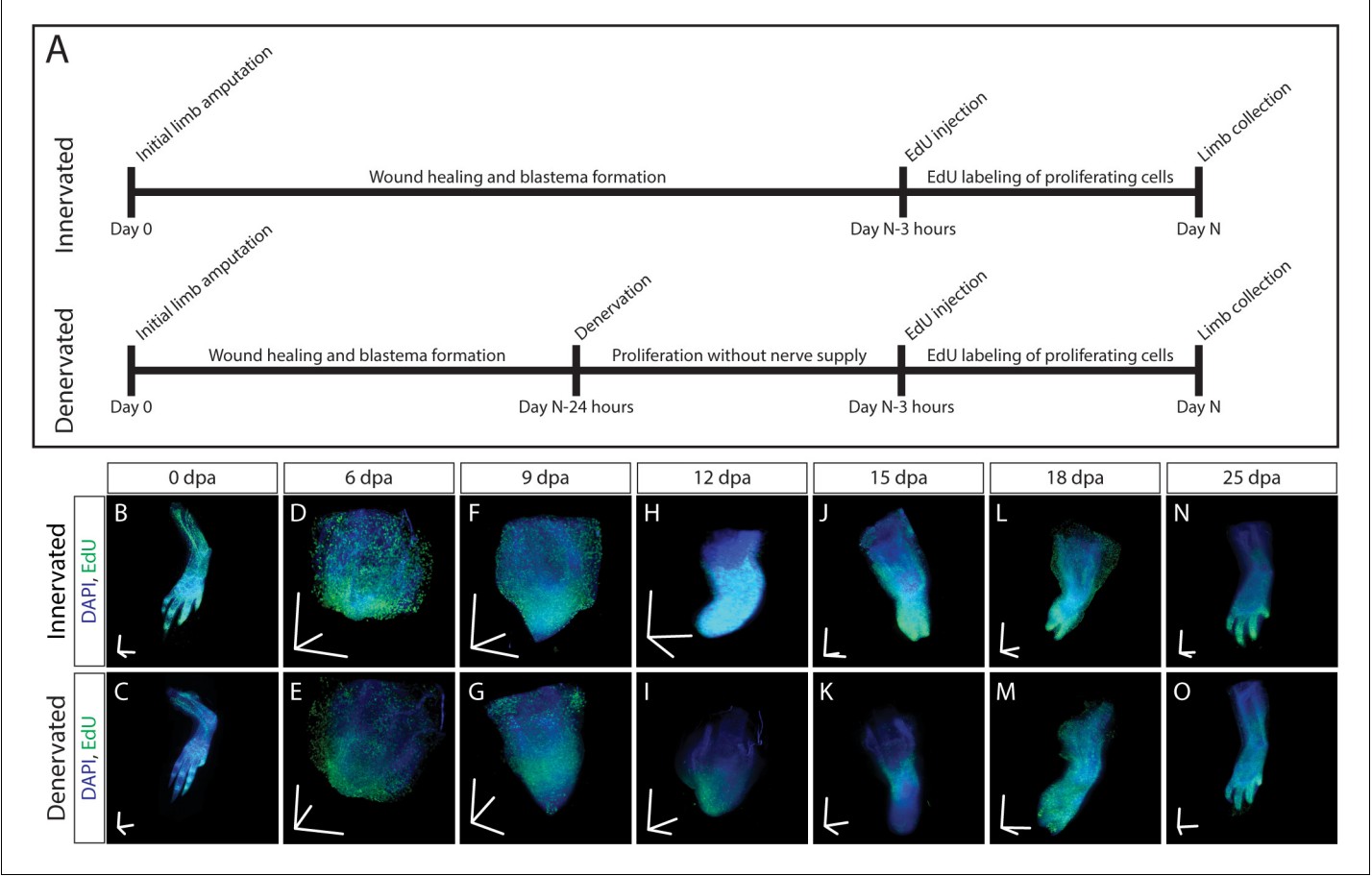

**Figure 5.** 3D visualization of DNA synthesis in innervated/denervated regenerating limbs. (**A**) Schematic of experimental design used to obtain samples from B-O. (**B–O**) Time course of regeneration in innervated and 24 hr denervated limbs at 0, 6, 9, 12, 15, 18, and 25 dpa. Scale bars for panels B-O = 600 μm for each axis.

incorporation to provide a snapshot of DNA synthesis in the 30 hr total S phase of blastema cells. LSFM was used to image samples (*Figure 5B*), ensuring pixel resolution was consistent between samples. We quantified DNA synthesis in denervated limbs compared to innervated limbs by creating a 175 × 175 × 175 μm cube 250 μm from the distal most tip of the blastema (*Figure 6A*). The size and location of the quantification cube was chosen to maximize the number of EdU⁺ mesenchymal cells and to exclude any epithelial cells. Although we limited the cube to the size of the smallest blastema, these parameters can be customized depending on the size of the blastemas. For this study, we estimate 50–150 blastema cells are found within the cube. From these results, we observed a marked decrease in blastema EdU incorporation due to denervation at 9, 12, and 15dpa (*Figure 6B*, *Figure 6—figure supplement 1*), demonstrating that our whole-mount staining approach is capable of detecting changes in macromolecule synthesis after biological perturbations. One potential limitation of our method in the blastema is the inability to perform single cell segmentation. This is due to the density and abundance of cells within the blastema. We predict that higher magnification imaging and deep learning segmentation techniques may overcome this limitation but will significantly increase imaging time and file size. For comparison, we have included estimates of imaging time and file sizes for confocal microscopy with a 20X objective and LSFM with a 5× objective (*Supplementary file 1*).

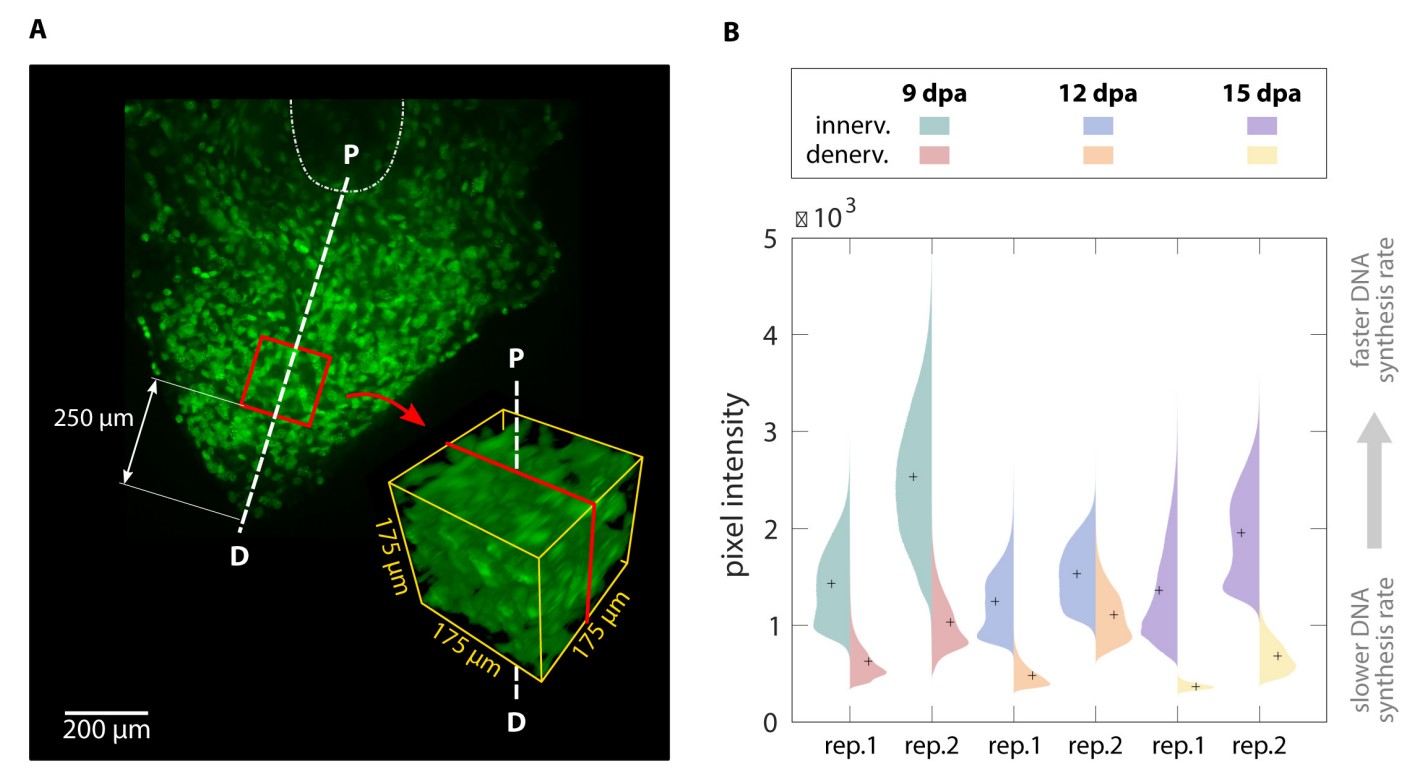

**Figure 6.** 3D quantification of DNA synthesis in innervated/denervated regenerating limbs. (**A**) A cube with sides of 175 µm was cropped along the proximodistal axis 250 µm from the distal tip of each blastema. P = proximal, D = distal. (**B**) Violin plots illustrate the pixel intensity of the innervated vs denervated blastema cubes. Comparison of mean intensity values (marked with a cross) of the same animal confirms that innervated blastemas have faster DNA synthesis rates than their denervated counterparts.

The online version of this article includes the following figure supplement(s) for figure 6:

**Figure supplement 1.** Pixel intensity histograms from innervated and denervated limbs.

## Discussion

The work presented here provides a fast, simple pipeline for visualizing macromolecule turnover in the 3D space of whole tissues. While tissue sections were previously the standard for studying these cellular processes, a new standard in the field must be expected where dynamic processes like macromolecule synthesis are visualized in 3D to obtain a more complete understanding of how these processes occur in a larger tissue context. Few modalities of imaging exist to provide this level of analysis. Here, we outline a method to study macromolecule synthesis at the organ, cellular, and molecular levels of organization, which is important in understanding cell state and how cell state affects neighboring cells and tissues. To this end, we show high levels of DNA synthesis, transcription, translation, and protein glycosylation in the entire 3D space of the regenerating blastema after limb amputation and that these processes can be visualized concurrently. Our lab has also demonstrated this whole-mount technique in other axolotl tissues including the lung (*Jensen, 2018*). We outline a multiscale pipeline for analysis and quantification of heterogeneous tissues at the organ, cellular, and molecular levels of organization. Further, we demonstrate that our method is sensitive to detect biological perturbations by showing a decrease in DNA synthesis in the blastema following limb denervation.

We foresee our method being used to similarly readdress other classical questions with modern techniques for a more exhaustive understanding of biological processes; traditional questions within the fields of cancer biology and neurobiology may especially benefit from technology as such. Additionally, as more click-it ready macromolecules monomer analogs are generated, our method will

provide a means to study more biological processes in whole tissues. Finally, we expect that this method should be amenable both with other staining techniques such as whole-mount immunohistochemistry and to a number of animal models, including mouse and zebrafish.

# Materials and methods

## Key resources table

| Reagent type (species) or resource | Designation | Source or reference | Identifiers | Additional information |
|---|---|---|---|---|
| Strain, strain background (*Ambystoma mexicanum*) | d/d axolotl | Ambystoma genetic stock center | RRID:AGSC_101L | |
| Commercial assay, kit | 5-Ethynyl-2′-deoxyuridine (EdU) | clickchemistrytools.com | Cat# 1149 | Monomer analog |
| Commercial assay, kit | 5-Ethynyl Uridine (5-EU) | clickchemistrytools.com | Cat# 1261 | Monomer analog |
| Commercial assay, kit | L-Azidohomoalanine (AHA) | clickchemistrytools.com | Cat# 1066 | Monomer analog |
| Commercial assay, kit | N-azidoacetylglucosamine-tetraacylated (Ac4GlcNAz) | clickchemistrytools.com | Cat# 1085 | Monomer analog |
| Commercial assay, kit | AFDye 488 Azide | clickchemistrytools.com | Cat# 1275 | Fluorescent azide |
| Commercial assay, kit | AFDye 594 Azide | clickchemistrytools.com | Cat# 1295 | Fluorescent azide |
| Commercial assay, kit | AFDye 594 Alkyne | clickchemistrytools.com | Cat# 1297 | Fluorescent alkyne |
| Chemical compound, drug | 2,2′-Thiodiethanol (TDE) | Sigma | Cat# 166782 | Clearing agent |
| Chemical compound, drug | L(+)-Ascorbic acid sodium salt | Thermo | Cat# AC352680050 | Click-it cocktail component |
| Chemical compound, drug | Copper(II) sulfate pentahydrate, 98+% | Thermo | Cat# AC423615000 | Click-it cocktail component |
| Chemical compound, drug | SlowFade Gold Antifade Mountant | Thermo | Cat# S36936 | Mountant |

## Animal procedures

Axolotls were either bred in captivity at Northeastern University or purchased from the Ambystoma Genetic Stock Center at the University of Kentucky. Experiments were performed in accordance with the Northeastern University Institutional Animal Care and Use Committee. Animals were grown to 4–6 cm (Mean 5.3 cm, SD 0.36) and 1–1.5 g (Mean 1.3 g, SD 0.19 g) for use in all studies. For all experiments, animals were anesthetized by treatment of 0.01% benzocaine until visually immobilized. Limbs were amputated either at the distal end of the zeugopod or midway through the stylopod, and bones were trimmed below the amputation plane to allow for uniform growth. At the date of collection, animals were reanesthetized and injected with either 5-ethynyl-2′-deoxyuridine (EdU) to identify proliferating cells (8.0 μg/g animal), 5-Ethynyl Uridine (5-EU) to label RNA (270.0 μg/g animal), L-Azidohomoalanine (AHA) to label protein (180.59 μg/g animal), or N-azidoacetylglucosamine-tetraacylated (GlcNAz) to label glycosylated proteins (430 μg/g animal) alone or simultaneously in the following combinations: EdU/AHA, EdU/GlcNAz, 5-EU/AHA, 5-EU/GlcNAz (*Table 1*). All monomer analogs were purchased from www.clickchemistrytools.com and resuspended in DMSO at the following concentrations: EdU- 300 mM, 5-EU- 100 mM, AHA- 100 mM, GlcNAz- 100 mM. Stocks were further diluted in 1× phosphate buffered saline (PBS) for injection. After 3hr of analog incorporation, limbs were collected from the upper stylopod and fixed in 4% paraformaldehyde (PFA) (diluted in 1× PBS) at 4°C overnight. If limbs were denervated, the nerve supply was severed at the brachial plexus 24 hr before tissue collection.

## Whole-mount click-it protocol

Following fixation in 4% PFA, samples were washed three times with 1X PBS at room temperature (~23°C) for 5 min. Samples were dehydrated in an increasing methanol series at room temperature starting with 25% methanol (diluted in 1× PBS), 50% methanol, 75% methanol, and 100% methanol for 5 min at each step. Samples could then be stored in 100% methanol indefinitely at −20°C. For staining, samples were rehydrated in a decreasing methanol series starting with 75% methanol (diluted in 1× PBS), 50% methanol, 25% methanol, and finally placed in 100% 1× PBS for 5 min at each step. Samples were then washed three times with 1× PBST (1× PBS with 0.1% Triton) for 5 min at room temperature. To aid in clearing, samples were washed in 0.5% trypsin (diluted in 1× PBS) for 30–90 min on a rocker at room temperature, or until the sample appeared translucent. Samples were washed three times at room temperature for 5 min with deionized water, then washed in 100% acetone for 20 min at −20°C and washed with deionized water again for 10 min. Samples were washed in 1× PBST three times at room temperature for 5 min before applying click-it cocktail for overnight at room temperature. The click-it cocktail was made in 500 µl of 1× TRIS buffered saline as follows: 50 µL 1M sodium ascorbate (100 mM final), 20 µL 100 mM $CuSO_4$ (4 mM final), and 2 µL 500 µM azide- or alkyne-modified Alexa Flour (2 µM final), combined in order as listed. After the first round of staining, samples were washed at room temperature six times for 30 min with rocking. For double-labelling, samples were again placed in the click-it cocktail with a different fluorescent dye to stain for the second analog at room temperature overnight. Both rounds of staining were conducted in the dark to prevent photodegradation of fluorescent molecules. For staining with DAPI, samples were washed three times for 5 min with 1× PBS, then placed in 2.86 µM DAPI for 4 d at room temperature. Samples were washed three times for 20 min with 1× PBS and left in 1× PBS at 4°C for short-term storage before imaging with LSFM. whole-mount samples were cleared with 67% TDE (diluted in 1× PBS) overnight at room temperature in the dark.

## Tissue section click-it protocol

Following fixation in 4% PFA, samples were washed three times in 1× PBS each for 5 min, and cryoprotected in 30% sucrose on a rocker until the tissue fully sank. Samples were removed form sucrose and briefly washed in optimal cutting temperature (OCT) compound before mounting in OCT compound and frozen at −80°C. A cryostat was used to obtain 10 µm sections, and slides were baked at 65°C for 15 min. Slides were then washed with water for 30 min at room temperature to remove residual OCT. Slides were washed once with 1× PBS for 5 min at room temperature. The click-it cocktail (same as above) was applied to the slides and incubated at room temperature in the dark for 30 min. If staining for a second macromolecule, slides were washed five times for 5 min with 1× PBS at room temperature. The samples were then stained for 30 min at room temperature in the dark using the above click-it cocktail with a different a fluorophore dye. Following the final click-it reaction, slides were washed once with 1× PBS at room temperature for 5 min, then stained with 2.86 µM DAPI for 5 min at room temperature. Slides were washed again with 1× PBS for 5 min at room temperature and water for 5 min at room temperature and mounted with SlowFade Gold Antifade Mountant. Slides were imaged using a Zeiss LSM800 confocal microscope.

## Light sheet microscopy

All 3D images were acquired using a Zeiss light sheet Z.1 microscope paired with Zen software. Unless otherwise indicated, samples were cleared and imaged in 67% TDE. Post-processing for visualization purposes was performed with Arivis Vision4D v3.1.4 on a workstation with a 64-bit Windows Embedded Standard operating system, and an Intel(R) Xeon(R) CPU E5-2620 v3 @ 2.40 GhZ (two processors), 128 GB RAM, and NVIDIA Quadro K2200 GPU. Sub-volumes were stitched together with the Tile Sorter in Arivis, using the manual projection option. Volume fusion was performed through automatic landmark registration of manually selected points for alignment. For visualization, background intensity was corrected using the automatic functionality.

## Data analysis

All data were processed on desktop computers with the aid of Fiji (*Schindelin et al., 2012*) and Matlab (*The MathWorks, 2019*). The custom Fiji scripts and Matlab codes used are available in the

supplementary material. We performed all analyses on unprocessed. czi files acquired directly from the light sheet microscope.

## Organ level

The goal of the organ level analysis was to determine gross morphology of the organ, such as area, principal axes, and maximum diameters in said axes for each cross section. Critical to this step is alignment of the images to a standard defined axis and reslicing in the transverse direction (perpendicular to the long axis of the humerus). The image stack of a regenerating axolotl elbow was imported into Fiji and aligned along the proximodistal axis of the humerus (*Figure 3A*). Morphological segmentation of the humerus was performed semi-automatically with the Segmentation Editor plugin (*Figure 3B*). The segmented surface was then exported from Fiji as a mesh via an. stl file using 3D Viewer (*Pietzsch et al., 2015*). Volumetric analysis was performed in Fiji by reslicing the aligned mask (*Figure 3B*) to obtain a stack of cross-sections perpendicular to the proximodistal axis, and then quantified the shape and volume with the Slice Geometry option in the BoneJ (*Doube et al., 2010*) Fiji plugin. Slice Geometry calculates cross-sectional geometric properties of shapes, including area, second moment of area around the major and minor axes, and maximum chord length from these major and minor axes.

## Cellular level

The goal of the cellular level analysis was to determine the number of proliferating cells (EdU$^+$) as a function of proximodistal position along the humerus, as well as average size, shape, and orientation. The aligned image stack (*Figure 3A*) was cropped using the morphological segmentation (*Figure 3B*) as a mask with the aid of the Image Calculator in Fiji. Based on the EdU staining (*Figure 3C*), we segmented the nuclei of the proliferating cells with Trainable Weka Segmentation 3D (*Arganda-Carreras et al., 2017*). We used a combination of filters available in Fiji before and after training the algorithm to improve segmentation results. Filters applied include the 3D Edge and Symmetry, Background Subtraction, 3D Fill Holes, Gaussian Blur 3D, and 3D Watershed Split. A detailed description of the sequence and parameters used is available in the supplementary materials. The 3D Objects Counter then provided a list of identified cells as well as their volume and position, among other information. In addition, the surfaces of the segmented cell nuclei were exported and processed in MeshLab, similarly to the process followed with the organ segmentation, to be visualized in Matlab (*Figure 3E*). The centroid coordinates and corresponding object volumes identified in Fiji were imported into Matlab for further processing and plotting. We computed the number of cell centroids within 50 µm thick slices perpendicular to the proximodistal axis to obtain a density-like measure of proliferating cells. Slice thickness was selected as slightly larger than the average cell diameter to ensure cells were not sampled across more than two slices.

## Molecular level

The goal of the molecular level analysis was to determine molecular activity, characterized by fluorescent signal, which in our study represented DNA replication (EdU). The aligned and cropped image stack used as a starting point of the cellular-level analysis (*Figure 3C*) was also analyzed at the molecular level. We resliced the image stack along the proximodistal axis in the organ under study (*Figure 3F*). The histogram and pixel intensity statistics were listed for each slice via the getRawstatistics function. These allow for further quantification of the fluorescent signal in Matlab, for example, we calculated the mean intensity of each plane along the proximodistal axis and the histogram of the whole organ. For the example showing nerve-dependent regeneration in axolotl blastemas (*Figure 6A*), we compared overall pixel intensity of the left (denervated) and right (control, innervated) forelimbs of the same animal. To ensure full repeatability of our data analysis, we selected the same cubic volume in all blastemas processed: a cube with 175 µm sides, centered along the proximal-distal axis and at a distance of 250 µm from the distal tip (*Figure 6A*). The cube size was adjusted to maximize the cube volume for all blastemas processed, while ensuring that the entire cube was contained within the blastema. Histograms of each cubic volume and the ratio between the innervated and denervated limb of the same animal were computed and plotted with Matlab (*Figure 6B*) to quantify the changes in proliferation in innervated versus denervated limbs at

different stages of regeneration. All scripts for blastema cube quantification are provided in the supplementary materials.

## Acknowledgements

The authors would like to acknowledge Tyler Jensen for his early assistance in developing the whole-mount staining protocol, and both Alex Lovely and Gouxin Rong for their microscopy and image analysis assistance. Images were obtained from the Harvard University Center for Biological Imaging and the Northeastern University Chemical Imaging of Living Systems core. We thank the Institute for Chemical Imaging of Living Systems at Northeastern University for consultation and imaging support. We acknowledge animal support from the Ambystoma Genetic Stock Center funded by NIH grant P40-OD019794.

## Additional information

### Funding

| Funder | Grant reference number | Author |
| --- | --- | --- |
| National Science Foundation | 1727518 | Sandra J Shefelbine |
| Northeastern University | Matz Scholarship | Eun Kyung Jeon |
| Northeastern University | Undergraduate Research Fellowship | Eun Kyung Jeon |
| National Science Foundation | 1656429 | James R Monaghan |
| National Science Foundation | 1558017 | James R Monaghan |
| European Commission | MSCA-GF 841047 CompLimb | Ester Comellas |

The funders had no role in study design, data collection and interpretation, or the decision to submit the work for publication.

### Author contributions

Timothy J Duerr, Formal analysis, Investigation, Writing - original draft, Writing - review and editing; Ester Comellas, Data curation, Software, Formal analysis, Methodology, Writing - review and editing; Eun Kyung Jeon, Johanna E Farkas, Investigation, Methodology; Marylou Joetzjer, Julien Garnier, Methodology; Sandra J Shefelbine, James R Monaghan, Conceptualization, Supervision, Funding acquisition, Writing - original draft, Writing - review and editing

### Author ORCIDs

Timothy J Duerr  https://orcid.org/0000-0002-4945-0294
Ester Comellas  https://orcid.org/0000-0002-3981-2634
Johanna E Farkas  http://orcid.org/0000-0001-6540-7870
Sandra J Shefelbine  https://orcid.org/0000-0002-4011-9044
James R Monaghan  https://orcid.org/0000-0002-6689-6108

### Ethics

Animal experimentation: Axolotls (Ambystoma mexicanum: d/d RRID Catalog #101L) were either bred in captivity at Northeastern University or purchased from the Ambystoma Genetic Stock Center at the University of Kentucky. Experiments were performed in accordance with Northeastern University Institutional Animal Care and Use Committee. Animals were grown to 4-6cm (Mean 5.3cm, SD 0.36) and 1-1.5g (Mean 1.3g, SD 0.19g) for use in all studies. For all experiments, animals were anesthetized by treatment of 0.01% benzocaine until visually immobilized.

### Decision letter and Author response

Decision letter https://doi.org/10.7554/eLife.60354.sa1

Author response https://doi.org/10.7554/eLife.60354.sa2

## Additional files

### Supplementary files

• Source code 1. Source code for multiscale analysis in *Figures 3* and *4*. Source code used in the 3D quantification across scales of a regenerating axolotl humerus depicted in *Figures 3* and *4*. A tutorial describing the process in detail is provided together with the ImageJ and Matlab scripts. This material is also available on Zenodo at https://doi.org/10.5281/zenodo.3891878. The original image used as the starting point is available at Northeastern University's Digital Repository and also upon request to the authors.

• Source code 2. Source code for DNA synthesis analysis in *Figure 6*. Source code used in the 3D quantification of DNA synthesis in innervated/denervated regenerating limbs depicted in *Figure 6*. Annotated ImageJ and Matlab scripts are provided. The raw images are available at Northeastern University's Digital Repository and also upon request to the authors.

• Supplementary file 1. Imaging parameters for LSFM and confocal microscopy. Key imaging parameters for imaging whole tissues with a 5× objective in LSFM, or imaging 10 µm tissue sections with a 20× objective in confocal microscopy.

• Transparent reporting form

### Data availability

Original data and scripts used to process the data are available at Northeastern University's Digital Repository http://hdl.handle.net/2047/D20388308.

The following dataset was generated:

| Author(s) | Year | Dataset title | Dataset URL | Database and Identifier |
|---|---|---|---|---|
| Duerr TJ, Comellas E, Jeon EK, Farkas JE, Joetzjer M, Garnier J, Shefelbine SJ, Monaghan JR | 2020 | 3D Visualization of Macromolecule Synthesis | http://hdl.handle.net/2047/D20388308 | 3D Visualization of Macromolecule Synthesis, D20388308 |

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
