## [Decision Letter]

**Acceptance summary:**

This study presents a combination of techniques to examine macromolecule synthesis in a complex and rapidly changing structure such as the regenerating limb of an axolotl. A pipeline for 3D visualization and quantification reveals the heterogenous cellular response within a complex tissue. Of interest, is the compatibility to use in other organs and tissues or at the organismal level.

**Decision letter after peer review:**

Thank you for submitting your article "3D Visualization of Macromolecule Synthesis" for consideration by *eLife*. Your article has been reviewed by three peer reviewers, including Tatiana Sandoval-Guzman as the Reviewing Editor and Reviewer #1, and the evaluation has been overseen by Didier Stainier as the Senior Editor.

The reviewers have discussed the reviews with one another and the Reviewing Editor has drafted this decision to help you prepare a revised submission. This decision includes a summary redacted from the three reviews.

Summary:

Duerr and colleagues present a combination of click-it, tissue clearing and light sheet technologies to examine macromolecule synthesis (namely DNA synthesis, transcription, translation, and glycosylation). This pipeline allows the visualization and quantification of cellular processes in growing 3-dimensional tissue. The authors use axolotl limb regeneration, a challenging system to tackle given its complex tissue types and large volume. Upon amputation, the remaining tissue in the limb will produce proliferating progenitors that accumulate forming a structure called the blastema, which the authors use to test macromolecule synthesis simultaneously. The authors demonstrate that administration of macromolecule analogs can be visualized simultaneously in a single sample. Furthermore, the authors create an image analysis pipeline that allows the analysis of individual elements or segments within the limb; they use the regenerating humerus to demonstrate the multiscale quantitative analysis capabilities of their method.

While the technologies used in this manuscript are not novel, the combination of these techniques together provide a compelling foundation to answer questions in this and other similar systems, while decreasing labor-intensive techniques such as tissue sectioning. The ability to look into these processes while preserving structure, helps understand the underlying biological function of a cell and its intricate relation to other tissues. This method opens up the possibility to analyze complex heterogeneous changes across heterogeneous tissues.

Overall, this manuscript has a potential broad scope and by using already commercial and open source tools, it can facilitate a broader applicability.

We have the following suggestions to improve the validation of the methods described in this manuscript.

Essential revisions:

1) One of the strengths of this methods is the diminution of a labor-intensive sectioning/staining/imaging track. One concern is that a more persuasive effort could be presented to validate the number of cells identified by both methods. Using the EdU+ quantification as a point of reference between the imaging protocols, the authors could provide a direct comparison with the standard methodology. One possibility is a randomized quantification of single planes from the wholemount volume and tissue sections. This would demonstrate side by side the detection sensitivity of the whole mount/LSFM visualization. Additionally, there is no information related to imaging parameters such as magnification, pixel size and its comparison between wholemount light sheet and confocal imaging. Many of the figures would be greatly enhanced by zoom insets to show the level of resolution and a comparison of resolution between the two methods. Can the same conclusion be reached using two-dimensional section imaging as it can using 3-dimensional imaging?

2) Given the complex nature of a regenerating limb, we consider of importance the biology behind the methods, more information is needed describing the rationale for the given experimental choices. In general, there is very little information shown for the methods. For example, macromolecule analogs were injected three hours before tissue harvesting. Why was this timepoint chosen? Is this time choice relevant for the additional analogs AHA or GlyNAz? Is there an optimization that the authors didn't mentioned? Similarly, the authors do not explain the use of TDE as a clearing agent versus other clearing strategies that they highlight in the manuscript. Is there an advantage or reason for this choice? The authors should clearly state if there is.

3) A key point of the paper is the simultaneous visualization and quantification of macromolecules. While the authors demonstrate that co-administration of macromolecule analogs is possible for simultaneous visualization, the further analysis doesn't exploit this important feature. We wonder if this method allows for identification of cells through immunofluorescence/immunohistochemistry together with the click-it macromolecule labeling. Did the authors attempt to immunolocalize a particular cell population? One particular question that we would like to see addressed is the quantification of RNA or protein synthesis in double labelled tissue (in volume). The authors may already have tissue that is double labeled.

4) How applicable can this method be for other tissues/organisms, or is it limited to more dense tissue (bone)? Have the authors attempted this technique on other organs? If it is envisioned to replace tissue sectioning, the authors should address whether the technique is adaptable to other tissues and model organisms. If generalizable, the associated supplementary tools and tutorial should also reflect this (e.g., the ImageJ macro currently prompts for a "blastema file").

5) The methods and rationale for the limb denervation experiments should be better explained. Specifically, why was a mock denervation not needed? In the absence of this control, could the change in DNA synthesis be due to the injury of the denervation, rather than an absence of nerves? Please also include more details on the time point chosen to denervate the limb, how the time point analyzed (24 hrs post denervation) relates to the timing of nerve degeneration and why such a small volume was analyzed. One possibility is that a small volume could bias the final histogram measurement. Did the authors consider analyzing two volumes in the blastema, for example, one proximal and one distal?

6) Figure 6 can show the power of the method, however, there is no statistical test to quantify the difference between the two experimental groups. This computational pipeline is giving numerical values to a biological process, it is worth then to analyse them. The graph in Figure 6 shows only the histograms of the 3 time points that the authors considered affected. It would be informative to know how the time points that show no change look in comparison. In general, it is challenging to interpret Figure 6, we suggest separating the histograms, or using violin plots for better visualization.

---

## [Author Response]

Essential revisions:1) One of the strengths of this methods is the diminution of a labor-intensive sectioning/staining/imaging track. One concern is that a more persuasive effort could be presented to validate the number of cells identified by both methods. Using the EdU+ quantification as a point of reference between the imaging protocols, the authors could provide a direct comparison with the standard methodology. One possibility is a randomized quantification of single planes from the wholemount volume and tissue sections. This would demonstrate side by side the detection sensitivity of the whole mount/LSFM visualization. Additionally, there is no information related to imaging parameters such as magnification, pixel size and its comparison between wholemount light sheet and confocal imaging. Many of the figures would be greatly enhanced by zoom insets to show the level of resolution and a comparison of resolution between the two methods. Can the same conclusion be reached using two-dimensional section imaging as it can using 3-dimensional imaging?

We have included estimates of cell counts in LSFM vs. confocal images. The line reads “Furthermore, the number of EdU+ cells per blastema area in a single Z slice obtained by LSFM (EdU-AHA: 1954 cells/mm^2^, EdU-GlcNAz: 1458 cells/mm^2^) is similar to that of a tissue section obtained by confocal microscopy (EdU-AHA: 1272 cells/mm^2^, EdU-GlcNAz: 1349 cells/mm^2^).”

We have included Supplementary file 1 with imaging parameters for LSFM and confocal imaging.

We have included Figure 2—figure supplement 3 to show the resolution that can be obtained with empty magnification of a picture obtained with LSFM.

2) Given the complex nature of a regenerating limb, we consider of importance the biology behind the methods, more information is needed describing the rationale for the given experimental choices. In general, there is very little information shown for the methods. For example, macromolecule analogs were injected three hours before tissue harvesting. Why was this timepoint chosen? Is this time choice relevant for the additional analogs AHA or GlyNAz? Is there an optimization that the authors didn't mentioned? Similarly, the authors do not explain the use of TDE as a clearing agent versus other clearing strategies that they highlight in the manuscript. Is there an advantage or reason for this choice? The authors should clearly state if there is.

Our goal for the pulse labelling was to obtain the strongest signal, while providing information on only the newest macromolecules. We have found that strong labelling occurs at 3 hours of labeling for EdU. We have observed from previous experience that 1 hour EU labeling provides a weak signal while 3 hours provides a signal comparable to a 3 hour EdU pulse. For consistency with these results, we presumed that 3 hours of AHA and GlcNaz labeling would also provide strong labeling of translation and glycosylation in action. We clarified this information in the text. The lines read “Our group has found that collection of tissue after three hours provides sufficiently strong metabolic labeling of DNA synthesis (EdU), transcription (EU), translation (AHA), and protein glycosylation (GlcNAz).”

We chose TDE based upon its simplicity, cost, safety, and compatibility with common imaging objectives. We include our rationale for using TDE as our clearing reagent. The lines read “TDE was chosen due to its simplicity, cost, safety, and compatibility with common imaging modalities.”

3) A key point of the paper is the simultaneous visualization and quantification of macromolecules. While the authors demonstrate that co-administration of macromolecule analogs is possible for simultaneous visualization, the further analysis doesn't exploit this important feature. We wonder if this method allows for identification of cells through immunofluorescence/immunohistochemistry together with the click-it macromolecule labeling. Did the authors attempt to immunolocalize a particular cell population? One particular question that we would like to see addressed is the quantification of RNA or protein synthesis in double labelled tissue (in volume). The authors may already have tissue that is double labeled.

We suspect that our method would be compatible with immunohistochemistry. However, whole mount immunohistochemisty is still a developing technology in the axolotl. Our labs are working to optimize this technology, but for the purpose of this paper we wished to demonstrate a method for which macromolecule synthesis can be visualized and quantified in 3D.

Quantification of macromolecule synthesis in double labelled tissue is an important question. We suspect that the method we developed for quantifying pixel intensity within a 3D cube would be sufficient to also quantify nascent RNA and protein synthesis in the blastema. Unfortunately, we do not have tissue ready for imaging, but plan on running these experiments in the future. Given the circumstances surrounding the COVID-19 pandemic with limited people in our laboratory and limited microscope availability, it would be difficult for this experiment to be completed by the time revisions are due.

We added the following lines in the Discussion to address compatibility with these methods, “Finally, we expect that this method should be amenable both with other staining techniques such as whole mount immunohistochemistry and to a number of animal models, including mouse and zebrafish.”

4) How applicable can this method be for other tissues/organisms, or is it limited to more dense tissue (bone)? Have the authors attempted this technique on other organs? If it is envisioned to replace tissue sectioning, the authors should address whether the technique is adaptable to other tissues and model organisms. If generalizable, the associated supplementary tools and tutorial should also reflect this (e.g., the ImageJ macro currently prompts for a "blastema file").

Yes, this method is amenable to a number of tissues. We show an example of staining entire larvae in Figure 1, which we now highlight explicitly in the manuscript. The lines now read “We show that our whole mount method can be used at the organismal level to visualize the whole torso of a stage 52 axolotl larvae, where we observe proliferating cells in the developing limbs and hematopoiesis within the spleen (Figure 1B).”

We also have a manuscript uploaded to BioRxiv that shows whole mount EdU staining of the axolotl lung. We now reference this manuscript in the Discussion. The lines now read “Our lab has also demonstrated this whole mount technique in other axolotl tissues including the lung (Jensen et al., 2018).”

We have modified the ImageJ macro for generalizable use.

5) The methods and rationale for the limb denervation experiments should be better explained. Specifically, why was a mock denervation not needed? In the absence of this control, could the change in DNA synthesis be due to the injury of the denervation, rather than an absence of nerves? Please also include more details on the time point chosen to denervate the limb, how the time point analyzed (24 hrs post denervation) relates to the timing of nerve degeneration and why such a small volume was analyzed. One possibility is that a small volume could bias the final histogram measurement. Did the authors consider analyzing two volumes in the blastema, for example, one proximal and one distal?

The surgery for a mock denervation involves opening a small area of epithelium in the axilla of the limb. Once this area is opened, the nerves can be accessed and cut. Johnson et al. 2018 (DOI: 10.1016/j.ydbio.2017.07.010) showed that skin injuries on the limb have little impact on EdU incorporation. For this reason, we deemed it unnecessary to use a mock denervation. This has also not generally been the standard in limb denervation studies in the past, although we see the utility for rigorous experimental design in the future. Our assumptions are based upon a series of experiments from Marcus Singer between 1942-1950 showing it is the absence of a sufficient motor, sensory, and sympathetic innervation that supports proliferation during regeneration and not an injury alone. As an example, in 1943 Singer cut the dorsal and ventral roots of dorsal root ganglia 3, 4, and 5 and showed that limbs still regenerated comparable to contralateral control amputated limbs. This demonstrates that as long as sufficient nerve axons reach the limb, it will regenerate. It is the lack of these axons in the limb after denervation that leads to a lack of proliferation. As examples of a lack of sham control through the years using salamander denervation, see Butler and Schotte, 1941 (Histological alterations in denervated non-regenerating limbs of urodele larvae); Olsen et al., 1984 (Rescue of blocked cells by reinnervation in denervated forelimb stumps of larval Ambystoma); Wu et al., 2013 (de novo transcriptome sequencing of axolotl blastema for identification of differentially expressed genes during limb regeneration). We agree with the reviewer though that this should be a standard control in the field moving forward.

More information on the 24-hour denervation timepoint was included in the text. The lines read “Blastema cells are thought to have a cell cycle length of 40-50 hours with S phase approximately 30 hours (Tassava, Goldhamer and Tomlinson, 1987). […] Three hours of EdU incorporation was chosen to provide a snapshot of DNA synthesis in the 30 hour total S phase of blastema cells.”

More information on the size/location of the quantification cube was included in the text. We estimate that 50-150 blastema cells are located within the quantification cube. In terms of the cube that was measured in limbs, we generated the largest cube possible to fit within all blastemas at all time points. Larger fields of view could be chosen in later blastema stages, but for comparison across time points, we decided to limit our analysis to the largest cube available across all limbs. Also, we chose to exclude the epithelium as this tissue has different cell cycle dynamics compared to the mesenchyme during regeneration. The new lines now read “The size and location of the quantification cube was chosen to maximize the number of EdU+ mesenchymal cells and to exclude any epithelial cells. […] For this study, we estimate 50-150 blastema cells are found within the cube.”

6) Figure 6 can show the power of the method, however, there is no statistical test to quantify the difference between the two experimental groups. This computational pipeline is giving numerical values to a biological process, it is worth then to analyse them. The graph in Figure 6 shows only the histograms of the 3 time points that the authors considered affected. It would be informative to know how the time points that show no change look in comparison. In general, it is challenging to interpret Figure 6, we suggest separating the histograms, or using violin plots for better visualization.

With a sample size of two in each group, we feel that a statistical test in this case would be of little consequence and not informative. With our multiscale analysis pipeline, we simply wished to demonstrate that macromolecule synthesis rates can be quantified using pixel intensity. This is also why we chose to analyze three time points. While we agree the biology would be interesting, we feel that the scope of our paper is to demonstrate the capabilities of our method to address biological questions. It is for this reason that we chose a well characterized biological perturbation such as denervation to demonstrate our method.

We now represent the data with violin plots in Figure 6 and it represents the data in a clearer manner. It is also obvious now after two biological replicates that the difference between innervated and denervated limbs is in accordance with the known impacts of limb denervation. We have also separated the histograms and included them in Figure 6—figure supplement 1.